# Perception and Attitude toward Teleconsultations among Different Healthcare Professionals in the Era of the COVID-19 Pandemic

**DOI:** 10.3390/ijerph191811532

**Published:** 2022-09-13

**Authors:** Urszula Grata-Borkowska, Mateusz Sobieski, Jarosław Drobnik, Ewa Fabich, Maria Magdalena Bujnowska-Fedak

**Affiliations:** 1Department of Family Medicine, Wroclaw Medical University, 51-141 Wroclaw, Poland; 2Department of Epidemiology and Health Education, Wroclaw Medical University, 50-372 Wroclaw, Poland; 3Jan Mikulicz-Radecki University Teaching Hospital, 50-556 Wroclaw, Poland

**Keywords:** healthcare professionals, teleconsultation, effectiveness, reliability, COVID-19 pandemic, doctor, nurse, midwife, paramedic, physiotherapist

## Abstract

Teleconsultation has become one of the most important and sometimes the only possible forms of communication between healthcare professionals (HCPs) and their patients during the COVID-19 pandemic. The perceptions and attitudes of HCPs to teleconsultations may affect the quality of the therapeutic process provided using them. Therefore, this study aimed to understand the attitudes to and perceptions of medical teleconsultation among various HCPs during the COVID-19 pandemic. We analyzed data from a dedicated questionnaire on preferences, attitudes, and opinions about teleconsultation, which was filled by 780 Polish HCPs. Most of the HCPs were doctors and nurses (69% and 19%, respectively); most of the doctors were family physicians (50.1%). During the pandemic, teleconsultation and face-to-face contact were reported as the preferred methods of providing medical services with similar frequency. Doctors and nurses displayed the most positive attitude toward teleconsultation while the paramedics and physiotherapists took the least positive view of it. The most frequently indicated ratio of the optimal number of teleconsultations to in-person visits in primary health facilities care was 20%:80%. Most HCPs appreciate the value of teleconsultation, and more than half of them are willing to continue this form of communication with the patient when necessary or desirable.

## 1. Introduction

Since the early 21st century, the world has seen a rapid development of information and communication technologies (ICT), which has affected many different areas of our life, including healthcare. Teleconsultation has become one of the most important, and sometimes the only possible, forms of communication between healthcare professionals and their patients during the COVID-19 pandemic. The pandemic has made it necessary to introduce solutions that would allow the patient to access remote medical services, which reduced the risk of coronavirus infection [1,2,3]. Teleconsultations have made it possible to avoid crowded waiting rooms, which has significantly reduced the risk of the spread of infection. Thus, telemedicine has made it possible to maintain the continuity of treatment, ensuring safe and timely provision of healthcare and reducing the costs of providing medical services [4].

The beginnings of Polish telemedicine date back to 1999 and the establishment of the Polish Society of Telemedicine. However, for several years, the development of telemedicine in Poland saw only slow progress. The breakthrough was the release of the Information System in Health Care Act in late 2015, which made it possible to provide medical services via telemedical systems, including by making diagnostic and therapeutic decisions [5]. Since then, many electronic functionalities have been implemented: sick leaves have been available online only since 2018, e-prescriptions since 2020, and e-referrals since 2021. Access to e-health services has also flourished elsewhere in the world: electronic prescriptions have been available in such places as Australia, Canada, New Zealand, the USA, and many European countries (Denmark, France, Germany, Sweden, Spain, UK); e-referrals have been used e.g., in Denmark, England, New Zealand, Norway, The Netherlands, and the USA [6,7]. While electronic sick leaves have seen less frequent use, we have found evidence of attempts to implement such a system in Germany, Qatar, Latvia, and Ukraine [8,9,10]. Thanks to the implementation of these new functionalities, teleconsultations have become an increasingly frequent form of contact between healthcare professionals (HCPs) and patients, especially among some specialists (e.g., psychiatrists, cardiologists, diabetologists), nurses, and midwives [11,12,13]. However, telecare used to be mostly concentrated on teleconsultations between primary care physicians and different specialists [14]. The real use of teleconsultations has exploded during the COVID-19 pandemic. For instance, it is estimated that the use of telemedicine solutions has doubled in the US Medicare program and has increased 30 times in the general US population in the second quarter of 2020 compared to the pre-pandemic period [15].

During remote visits, it is possible to monitor, consult, and inform patients about the diagnostic and therapeutic process, as well as educate them in their own environment. This contributes to improving patients’ quality of life [16,17,18]. Many factors may affect the reliability and effectiveness of the teleconsultation provided. One of the key factors in this regard is the proper preparation for this form of providing medical services, particularly through the collection of relevant data. Thanks to ICT solutions, HCPs and patients can share various types of information, including blood pressure test results, X-rays, and CT scans [19]. Remote access to patients’ medical history and the results of their medical tests enables the optimization of the therapeutic process and the establishment of a detailed care plan. From the technical standpoint, the availability of a stable internet/telephone connection is also vital, as is the possibility of ensuring privacy during teleconsultations [12,19].

Moreover, it should be noted that the quality of the diagnostic and therapeutic process is affected not only by technical factors but also aspects related to the HCPs’ and patients’ attitudes to providing medical services by teleconsultation. Choosing an appropriate approach for providing and using medical consultations in a brand-new way is all the more crucial during the COVID-19 pandemic when the frequency of teleconsultation has increased significantly. The studies conducted so far have shown that patient attitude has a significant impact on the quality of teleconsultations and the subsequent satisfaction with medical advice [20,21,22,23]. The attitude of HCPs toward using teleconsultations is also extremely important, enabling ever better adaptation to the dynamically changing conditions of the healthcare systems. However, there is a lack of research on perceptions and attitudes of HCPs in the broader context. During the literature review, we found only a few manuscripts that have focused mostly on a selected group of doctors [24,25]. Our study thus aimed to understand the attitudes and perceptions toward medical teleconsultations among various healthcare professionals during the COVID-19 pandemic.

## 2. Materials and Methods

To conduct the study, we created an original questionnaire that consisted of 11 questions concerning various aspects of the work of HCPs during the COVID-19 pandemic. The questions pertained to the preferred method of work (teleconsultation or personal visit), working conditions (availability of additional test results, duration of such visits), advantages and disadvantages of teleconsultation, and a subjective assessment of the effectiveness and reliability of teleconsultation. By “reliability” we understand the total subjective assessment of the possibility of carrying out a correct diagnostic process—the ease of collecting a medical interview, assessing the risk, verifying the patient’s ailments, or the possibility of deepening the case history. With the term “effectiveness” we understand the subjective assessment of the entire diagnostic and therapeutic process—i.e., both the speed and quality of conducting an anamnesis, providing feedback, implementing new recommendations, or controlling chronic diseases by means of teleconsultation. Participants in the study were instructed about the adopted definitions of these concepts, and in cases of doubt they were explained on the basis of these descriptions. Additionally, study participants were asked to fill in a metric about their sex, age, professional role, and specialization.

A translated version of the questionnaire used in the course of the study is available as Appendix A. Data were collected using both Google Forms sheets and paper versions of the questionnaires. A total of 1000 paper questionnaires were distributed to HCPs in primary healthcare clinics and hospitals in Poland’s Lower Silesia Province, receiving a rate of return of 36.8%. The electronic version of the questionnaire was sent to HCPs belonging to medical groups and associations (e.g., Polish Society of Family Medicine, Lower Silesian Chamber of Nurses, Midwives, etc.). We sent invitations to participate in the study to 4345 e-mail addresses (some of these addresses may have been duplicated if a given member belonged to several associations). Finally, 412 responses (52.8% of all responses received) were collected using an electronic version of the questionnaire, amounting to a rate of return of 9.48%. The questionnaires were collected between 9 April 2021 and 17 July 2021.

The Statistica program was used for data analysis. The Shapiro–Wilk normality test was applied to check the normal distribution using 0.05 as a significance level. Of all the variables studied, only two had no normal distribution: the assessment of the effectiveness and the assessment of the reliability of the teleconsultations performed; in both cases, the calculated p-value was less than 0.001. For this reason, non-parametric tests were used in the further study of these variables. The dependence of the above variables on the categories of qualitative variables was tested using the non-parametric Wilcoxon multiple comparisons test. This test enabled a direct comparison of the value of a quantitative variable between the two categories of a qualitative variable. In the case of qualitative variables, the uniformity of the category distribution was tested with the Chi-square uniformity test (one sample proportions test). The independence of the two qualitative variables was verified using Fisher’s exact test—this test was used instead of the Chi-square test due to the small number of observations in some of the categories. A significance level of 0.05 was set for all tests. In the case of some of the examined features (advantages and disadvantages of teleconsultation), only a descriptive interpretation was possible. This is mainly due to the inability to conduct further statistical analyses—these variables cannot be tested for homogeneity. The respondents could indicate several categories at the same time, so the categories do not meet the separability condition required for homogeneity testing.

## 3. Results

The analysis used data from 780 questionnaires on teleconsultation in healthcare collected from HCPs (31 questionnaires were rejected from the study due to missing answers, and 20 due to the fact that the participants did not provide medical services via teleconsultation yet completed the entire questionnaire). Data on study participants were displayed on a chart (Figure 1). The surveyed groups of healthcare professionals were broken down into doctors, nurses, physiotherapists, paramedics, and midwives. Among the surveyed professions, doctors accounted for the highest percentage (69.8%) while paramedics accounted for the lowest (3.2%). Most doctors were family doctors (50.1%), internists (14.9%), and pediatricians (7.1%). There were 577 women (74%) and 203 men (26%) among the respondents. The majority of the respondents belonged to the 31–40 age bracket (31.0%); the 60+ years old category was the least numerous (7.6%), as shown in Table 1.

Most Polish healthcare workers did not use teleconsultations in their work before the pandemic (69%), and the use of teleconsultations at work before the pandemic depended significantly on their medical occupation (*p* < 0.001). Before the pandemic, 31% (n = 240) of HCPs used teleconsultations in their work—teleconsultations were most frequently used by midwives (60.7%) and least frequently by paramedics (8.0%). 

During the COVID-19 pandemic, there has been a similar preference for teleconsultations as for in-person visits—at 50.5% and 49.5%, respectively—but this depends on the professional group studied. Among doctors and nurses, the distribution of responses is similar yet among other professional groups the preference is higher for personal visits (*p* < 0.001), as shown in Table 2 and Appendix A.

Most healthcare professionals have access to the results of laboratory and imaging tests during teleconsultation—the most common answer to this question was “often” (49.9%); the least common ones were “rarely” (5.7%) and “I have no access” (8.9%). The HCPs’ ability to access test results during teleconsultation depends on their professional role (*p* < 0.001). Test results are most often available to doctors (“always”—22.7%, “often”—58.2%, “no”—3.1%) and the least often to physiotherapists (“always”—4.3%, “often”—21.7%, “no”—34.8%), as shown in Appendix A.

Most healthcare workers consider teleconsultation a time-saving method, with 47.8% of respondents stating that teleconsultation takes less time than a personal visit; 33.4% claimed that it takes the same amount of time. For 18.8% of respondents, teleconsultations are more time-consuming than in-person visits. The relationship between the time devoted to teleconsultation and a personal visit depends on age (*p* = 0.005). The percentage of healthcare workers who believe that the time needed for teleconsultations is shorter than that required for personal visits decreases with age—the youngest people, aged 20–30, most often point to a shorter teleconsultation time compared to a personal visit (61.0%); in contrast, those aged over 60 were the least likely to make that claim (41.5%). The relationship between the duration of a teleconsultation session and a personal visit also depends on the role performed (*p* < 0.001). Paramedics (70.8%) were the most likely to indicate that teleconsultation is a time-saving method compared to in-person visits. The necessity of devoting more time to a teleconsultation session than a personal visit was most commonly noted by physiotherapists (43.5%). The relation between the time used for teleconsultation and personal visits does not depend on the respondents’ sex or medical specialization (see Appendix A).

Four categories were most often indicated as the greatest advantages of teleconsultation during the COVID-19 pandemic: limiting one’s own possibility of contracting the SARS-CoV-2 virus (29.4%), reducing the risk for patients (28.4%), the possibility of providing advice to a larger number of patients (17.7%), and the possibility of quickly contacting the patient (18.0%). The ease of dealing with administrative matters, correction of chronic treatment, consultation of laboratory test results, and issuing of repeat prescriptions also proved a major advantage for a small percentage of respondents (3.3%).

The most frequently reported disadvantage of this medical consultation method was the inability to examine the patient personally (34.0%). Other frequently reported disadvantages included unreliable patient information and the inability to objectify it (22.4%), technical difficulties (19.5%), as well as difficulties resulting from the patient’s symptoms (e.g., deterioration of hearing, psychotic disorders) (22.5%)—see Table 3.

Most HCPs positively assessed the two basic aspects of teleconsultation—its effectiveness and reliability. The Likert scale was used to measure these values (from 1 to 10, where 1 is the lowest effectiveness/reliability and 10 is the highest). The average assessment of the effectiveness of teleconsultation was 6.19 (median 7), whereas reliability was assessed at a level of 5.89 (median 6)—see Appendix A.

The assessment of the effectiveness of teleconsultation depends on age—the youngest medical workers (20–30 years old) rated the effectiveness of teleconsultation higher than the oldest ones (over 60 years old), with the respective median ratings being 7 and 5 (*p* = 0.025). Similarly, slightly older medical workers (31–40 years old) rated the effectiveness of teleconsultation higher than the oldest workers (over 60 years old), with the corresponding median ratings being 7 and 5 (*p* = 0.019) as well—see Table 4. The assessment of the reliability of teleconsultation does not depend on age.

The evaluation of the effectiveness of teleconsultation depends on the respondent’s professional role. Doctors rated the effectiveness of teleconsultation higher than all other professions; the corresponding medians of ratings were 7 and 5 (*p* < 0.001), 7 and 5 (*p* = 0.008), 7 and 4 (*p* < 0.001), 7 and 3 (*p* < 0.001) for nurses, midwives, paramedics, and physiotherapists, respectively. Nurses and midwives rated the effectiveness of teleconsultation higher than physiotherapists, with the corresponding median ratings being 5 for nurses and midwives, and 3 for physiotherapists (*p* = 0.008 for nurses and physiotherapists; *p* = 0.026 for midwives and physiotherapists; see Table 4). The situation was similar for the reliability of teleconsultation—its assessment also depended on the profession. Doctors rated the reliability of teleconsultation higher than all other professions, with the corresponding median ratings of 7 and 5 (*p* < 0.001), 7 and 4 (*p* = 0.008), 7 and 3 (*p* < 0.001), 7 and 3 (*p* < 0.001) for nurses, midwives, physiotherapists, and paramedics, respectively. Nurses rated the reliability of teleconsultation higher than physiotherapists, with the corresponding median ratings being 5 and 3 (*p* = 0.018) for nurses and physiotherapists, respectively; the situation was similar in the case of midwives: the median ratings were 4 and 3 (*p* = 0.045) for nurses and midwives, respectively—see Table 5.

The assessment of the effectiveness and reliability of teleconsultation also depends on specialization—family medicine specialists rated the effectiveness of teleconsultation higher than other medical specialists: internists, with respective median ratings of 7 and 6 (*p* < 0.001), and cardiologists, with respective median ratings of 7 and 5 (*p* = 0.003). Similarly, family doctors rated the reliability of teleconsultation higher than doctors of other specializations (Table 4 and Table 5).

The distribution of answers to the question of whether the respondents intend to use the teleconsultation method in their professional work after the end of the COVID-19 pandemic was heterogeneous (*p* < 0.001). The most common answer was “yes, sporadically” (53.4%), the least frequent one was “I have no opinion” (4.2%). 

The willingness to use teleconsultation in everyday work after the COVID-19 pandemic depends on the age of HCPs (*p* = 0.020). The youngest people, those aged 20–30, most often declared a willingness to use teleconsultation after the pandemic ends (“yes, often”—35.8%, “no”—8.9%); people over 60 did so least often (“yes, often”—17.2%, “no”—27.6%). Overall, the willingness to use teleconsultation after the COVID-19 pandemic decreases with age. The willingness to use teleconsultation after the pandemic also depends on the occupation (*p* < 0.001). The readiness to use teleconsultation once the pandemic ends was most frequently declared by physicians (“yes, often”—34.8%, “no”—6.8%) and the least frequently by physiotherapists (“yes, often”—0%, “no”—19.2%). The willingness to use teleconsultation after the pandemic does not depend on sex (*p* = 0.083)—see Table 6.

Nevertheless, the majority of Polish healthcare professionals believe that most visits to primary healthcare facilities should be made in person. The most frequently indicated answer was that the ratio of personal visits to teleconsultations should be 20%:80% (25.2%). The assessment of the necessary ratio of teleconsultation to personal visits depends on the occupation (*p* = 0.005); doctors typically indicated the ratio of 20% teleconsultations to 80% of in-person visits, whereas nurses opted for a 30%:70% ratio, and physiotherapists and paramedics chose a 10%:90% ratio (see Appendix A).

## 4. Discussion

The COVID-19 pandemic has revolutionized the approach to telemedicine solutions and their application in everyday work in healthcare. It has increased the interest in telemedicine and forced the rapid implementation of teleconsultations by HCPs [20]. Our study found that less than a third of Polish HCPs had used teleconsultations before the pandemic. This is a smaller amount than that reported by other researchers; e.g., in Saudi Arabia, teleconsultation was used by more than half of the surveyed GPs [26]. Moreover, in the UK, as many as a quarter of the physician–patient interactions took place through teleconsultation [27]. On the other hand, this number exceeds that reported in a survey conducted among Brazilian doctors—less than 18.5% of them had used teleconsultation before the pandemic [28].

As mentioned in the introduction, the pre-pandemic teleconsultation activities in Poland were primarily aimed at organizing consultations of family doctors with other specialists. HCPs’ remote communication with patients was merely a niche addition to everyday work [14]. The exception, as our study showed, was midwives, since over 60% of them used teleconsultations to contact their patients before the COVID-19 pandemic. This is likely due to their close contact with puerperae during the pre- and postpartum periods when numerous simple questions arise. According to Pflugeisen and Mou, the usage of teleconsultation by midwives facilitates quick contact and clarification of doubts [13].

Our study indicates that the most frequently mentioned advantage of teleconsultations was the reduced risk of infection with the SARS-CoV-2 virus—for patients and HCPs alike. The aspect of reducing the risk of HCPs contracting the SARS-CoV-2 virus is of utmost importance in Polish healthcare conditions due to the high average age of HCPs, and hence, a higher risk of severe COVID-19 in the case of illness. The average age of doctors in Poland is 49.5 years, and one in four of them is over 65; as of 2014, the average age of midwives was 47.21, and that of nurses was 48.43, with a continuous upward trend [29,30]. The rapid aging of the medical workforce is also visible in other countries (e.g., Italy, France, Latvia, Israel, Hungary, and Belgium), as indicated in a report by the Organization for Economic Co-operation and Development [31]. It should also be mentioned that the fear of contracting the SARS-CoV-2 virus remains a vital reason behind the decrease in the number of patients reporting to hospitals and clinics [32,33]. Healthcare utilization decreased by about one-third during the COVID-19 pandemic, primarily among people with less severe illnesses [34]. Therefore, telemedicine makes it possible to maintain the continuity of treatment.

Telemedicine enables the provision of quick and effective medical services to patients, which was also indicated as an important advantage by 18% of our participants. For most of the respondents, providing teleconsultation is faster than on-premises patient admission, which is especially visible among younger HCPs. This is due to the higher technological proficiency of younger people, who find it easier to use new methods and tools [35,36]. Among the surveyed professional groups, it was the paramedics who most often pointed to the time savings enabled by teleconsultation. Thanks to remote contact, they can conduct an initial interview and thus decide whether or not to visit the patient at his or her home [37]. Time savings are less noticeable in other professional groups; physiotherapists gained the least extra time, which was due to the need to use direct contact methods at work or to demonstrate appropriate exercises, a task that may prove more difficult during remote contact [38].

Teleconsultations facilitate the safe, prompt, and continuous provision of healthcare—this was recognized by nearly one-fifth of our respondents. Remote contact makes it possible to prepare for the visit (e.g., after the initial triage) and facilitates the coordination of organizational matters, such as extending the validity of prescriptions [39]. Ease of access to HCPs is also crucial for patients, and teleconsultations make it possible to overcome some of the barriers they face (i.e., long travel time, difficulty in finding a parking space, impaired mobility or area exclusion, financial barriers, having to take a day off, or to care for another person) [39,40]. For children and adolescents, the use of teleconsultation reduces absenteeism from school [40]. Thanks to the numerous applications and electronic tools that enable remote contact with HCPs, even patients who have direct access to medical care can consider telemedicine a convenient alternative [41]. It is worth mentioning that remote contact makes it easier to provide psychological support, even when face-to-face contact is impossible. This is exceptionally important for patients with mental and psychiatric disorders, especially during the COVID-19 pandemic [42].

While telemedicine innovation has significant potential to benefit patients, it also presents various challenges. The disadvantages of telemedicine include the emergence of new threats to the quality, safety, and continuity of care, all of which can weaken the patient–physician relationship [41]. The threat that was most frequently reported by the study participants was the inability to physically examine the patient, which is consistent with the results of research by other authors [26,39,43]. This applies not only to a medical examination (e.g., temperature measurement, palpation, auscultation, or testing joint motion range) but also to a nursing examination (e.g., the nurse cannot touch the patient’s wound or recognize infection based on odor) [44]. The lack of in-person presence not only limits the physical examination possibilities but also prevents face-to-face contact. Over ninety percent of surveyed psychiatrists in Ireland complained about the inability to assess non-verbal communication during telephone calls, which prevents a reliable diagnosis of mental disorders [20]. The impossibility of verifying the reported complaints is strongly related to the issue of unreliable transmission of information by the patient, which was commonly reported by participants. Patients deliberately exaggerate or underestimate the seriousness of their symptoms, typically to obtain a sick leave or avoid being referred to the hospital [45]. Nevertheless, the safety of teleconsultation has been confirmed by a study conducted at the Onco-Hematology Center of Tor Vergata Hospital in Rome [46]. The study conducted by this center found that patients who received medical services through teleconsultation have returned for checkups after the re-opening outpatient departments, and no serious adverse events were reported. However, it should be borne in mind that the high effectiveness of providing medical services by means of teleconsultation in this hospital resulted from the implementation of internal protocols regarding patient privacy as well as regarding the method of examining the patient and obtaining data on the patient’s health using telemetry.

Another significant problem is the challenges in accessing technology faced by the elderly. Certain difficulties in using medical services were observed when trying to implement such a system among the elderly due to their limited access to equipment, or technical knowledge needed to participate in teleconsultations [26]. Today, the use of information and communication technologies is considered one of the fundamental aspects of citizenship [47]. The older patients’ inability to use these technologies leads to their digital exclusion. In Poland, over 56% of the elderly do not use the Internet; this percentage increases significantly the higher the age and the lower the education level [48]. This report shows that as many as 72.3% of seniors show a lack of interest in learning basic computer or Internet skills, and nearly two-thirds (65.1%) do not use the Internet for health-related matters. It is mostly younger people with a higher level of education who declare interest in an additional way to contact a doctor, apart from the traditional in-person visit, as well as in the possibility of remote health condition monitoring [49,50,51,52,53].

Another difficulty in conducting teleconsultations is the problems resulting from the patients’ chronic diseases. Hearing and vision impairment or mental disorders are particularly common in the geriatric population, which most often uses medical services. The largest screening study of health problems among the elderly in Poland (Project POLSENIOR) showed that almost 42% of them are affected by vision disorders, 10% suffer from hearing impairment, 15.8% show signs of dementia, and 7.8% declared having a stroke in their medical history, which is consistent with the worldwide research results on the frequency of such disorders [54,55,56]. For many patients in these groups, it is necessary to take additional steps to enable them to fully benefit from the advantages of teleconsultations.

Most of the surveyed participants positively assessed the reliability and effectiveness of teleconsultations. Higher effectiveness ratings were given by younger HCPs, probably due to greater skills in handling ICT service methods, which is consistent with the aforementioned research [35,36]. The effectiveness and reliability of teleconsultations were rated the highest by doctors, and the lowest by physiotherapists. Low perceived reliability of teleconsultations was also indicated by paramedics. The systematic review of the effectiveness of teleconsultation among primary care physicians, carried out by Carrillo de Albornoz et al., showed that teleconsultations and video consultations were as effective as face-to-face visits, and that patient satisfaction and compliance were high. However, the negative effect observed in this study was high discontinuation rates in patients receiving teleconsultations [57]. The effectiveness and safety of teleconsultation are also evident in the research carried out by other scientists among nurses and paramedics. Teleconsultation is a safe and time-saving method of management for “non-serious” emergency ambulance patients, and teleconsultations of paramedics with doctors remaining in the hospital did not extend the service provision time. [37,58,59]. Implementation of nurse triage for electronic consultation to improve access to specialty care reduces the waiting time for an appointment with a specialist and is a safe method of patient segregation [60]. Similar conclusions can be drawn from research on nurses’ telecare provided for patients with cardiovascular diseases—nursing teleconsultation has proven to be a method enabling continuity of care and outpatient management during the COVID-19 pandemic [61]. The positive effects of implementing teleconsultation are less evident among physiotherapists, although providing remote advice on how to perform appropriate exercises improves the physical functioning of patients [38].

The assessment of the reliability of teleconsultations in the research of other authors strictly depends on the area studied. For example, a meta-analysis on the reliability of teledermatology conducted by Bastola et al. brought highly divergent results [62]. In some studies, the reliability of teleconsultation was almost identical to on-premises visits. In other studies included in this meta-analysis, the reliability of both forms of medical services differed significantly in favor of inpatient visits. In contrast, a telemedicine system designed for rural Kenya made it possible to provide patients with approximately the same quality of care and advice as if the patient had physically traveled to a clinic. Such a positive conclusion seems to be particularly important in the case of remote places where contact with medical care is tremendously difficult [63].

During the COVID-19 pandemic, the distribution of received responses regarding the preferred method (teleconsultation vs. on-premises visits) was symmetrical, which shows the advantages of using teleconsultation during a health crisis. The vast majority of Polish HCPs intend to continue to use teleconsultation in their daily practice. This finding is close to the results obtained among primary care physicians in Saudi Arabia, where as many as 80% of respondents believed that telemedicine should continue to be used even after the pandemic [26]. However, almost all our respondents admit that the majority of visits should be conducted face-to-face. Doctors typically indicated an optimal ratio of 20% teleconsultations to 80% stationary visits; nurses pointed to a slightly higher ratio, while physiotherapists and paramedics opted for the lowest. This is because teleconsultations should be limited to cases where remote contact is completely sufficient for diagnosis and full, appropriate care for the patient.

There are some limitations to this study. The first is the small number of medical professionals other than doctors. This is because a large part of the data was collected using a questionnaire distributed via e-mail to members of various associations, and most of them were members of doctors’ associations, which might contribute to obtaining a decent response rate. Our attempts to increase the percentage of participants belonging to other professional groups proved unsuccessful due to the limited duration of the study. Yet another limitation is the heterogeneous sex distribution—this is probably because, statistically, women complete questionnaires and research surveys more often, and also because most Polish doctors, nurses, and midwives are female [64,65]. Another limiting factor is the lack of differentiation of respondents based on their technical proficiency or earlier usage of eHealth services other than teleconsultations, i.e., tele-auscultation or point-of-care devices transmitting the data to HCPs. It seems possible that higher information technology skills facilitate the use of teleconsultations by HCPs; however, on the basis of the collected data, it is impossible to confirm such a hypothesis. This issue can serve as inspiration for further research in the field of telemedicine. The last limitation is the non-homogenous distribution of age categories in the studied population—younger people are more willing to take part in scientific research, hence their percentage among all participants is higher [66].

## 5. Conclusions

Most HCPs appreciate the value of teleconsultations, and more than half of them are willing to continue this form of communication with the patient when necessary or desirable. The advantages most commonly reported by HCPs were the reduced risk of contracting the SARS-CoV-2 virus, quick contact with the patients, and the possibility of providing medical advice to more patients at the same time. Whether thanks to its advantages or despite its disadvantages, teleconsultation has improved access to high-quality, affordable care for patients during the ongoing pandemic while enabling social distancing at the same time. Teleconsultations have become an effective and reliable method of providing medical services. However, the attitude of Polish HCPs toward them varies depending on the tasks and specificity of individual medical professions. The reliability of teleconsultations was rated the highest by doctors and the lowest by physiotherapists and paramedics; their effectiveness was rated the highest by doctors and nurses and the lowest by paramedics.

One should bear in mind that telemedicine solutions cannot completely replace the personal, face-to-face interaction between HCPs and their patients, and are not an appropriate model of care for all conditions, e.g., those where direct physical examination is crucial, or important data can only be collected through face-to-face contact. It is also worth mentioning that the transition to the post-pandemic phase of using teleconsultation requires a key transformation of telehealth systems. There is a need to shift away from the crisis mode (where makeshift or untested technologies are allowed) to sustainable, secure systems that adequately preserve patient data security and privacy and offer sustained support.

## Figures and Tables

**Figure 1 ijerph-19-11532-f001:**
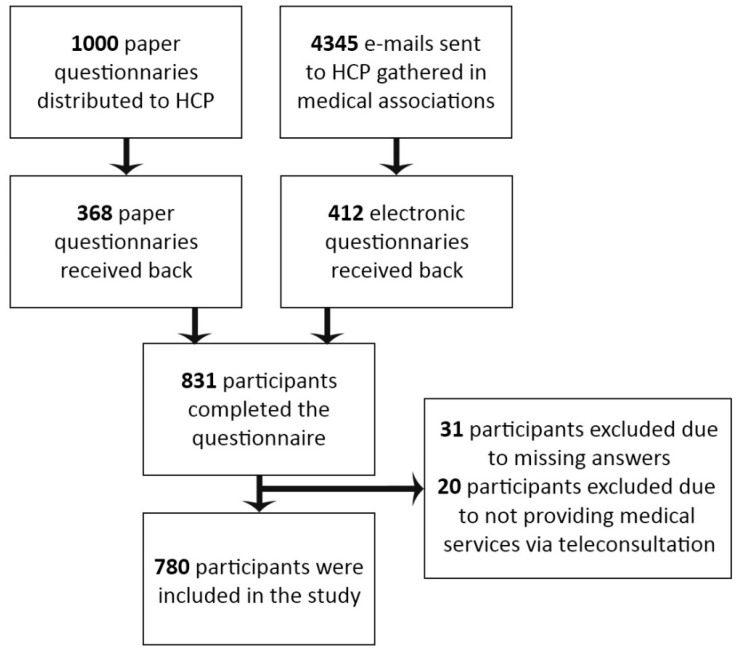
Flow chart showing the method of data collection and number of study participants.

**Table 1 ijerph-19-11532-t001:** Characteristics of participants in the study.

Variable	Categories	n	%	χ2 Cramer’s V	*df*	*p*
age	20–30 years of age	124	15.9	120.94	4	0
	31–40 years of age	242	31.0	0.197		
	41–50 years of age	185	23.7			
	51–60 years of age	170	21.8			
	>60 years of age	59	7.6			
sex	female	577	74.0	179.33	1	0
	male	203	26.0	0.480		
professional role	doctor	544	69.8	1282.02	4	0
	nurse	151	19.4	0.641		
	physiotherapist	31	4.0			
	paramedic	25	3.2			
	midwife	28	3.6			

Note. χ2—the homogeneity test, Cramers’ V—a coefficient determining the level of dependence between two nominal variables, df—number of degrees of freedom, *p*-calculated significance level (value 0 means *p* < 0.001).

**Table 2 ijerph-19-11532-t002:** Pre-pandemic work through teleconsultation and the preferred way of working during a pandemic among different professional groups.

	Using Teleconsultation before a Pandemic				Fisher’s Test Cramer’s V	Preferred Way of Working during the COVID-19 Pandemic				Fisher’s Test Cramer’s V
	Yes		No			Teleconsultation		Personal Visit		
	n	%	n	%	*p*	n	%	n	%	*p*
doctor	195	35.8	349	64.2	0 0.225	294	54.2	248	45.8	0 0.244
nurse	20	13.6	127	86.4		78	54.2	66	45.8	
physiotherapist	6	20.0	24	80.0		4	13.3	26	86.7	
paramedic	2	8.0	23	92.0		3	12.0	22	88.0	
midwife	17	60.7	11	39.3		9	32.1	19	67.9	

Note. *p*—calculated significance level in Fisher’s exact test of independence; (0 means *p* < 0.001), Cramers’ V—a coefficient determining the level of dependence between two nominal variables.

**Table 3 ijerph-19-11532-t003:** The advantages and disadvantages of teleconsultations among Polish HCPs.

Variable	Categories	n	%
Advantages of teleconsultation	reducing one’s own risk of contracting the SARS-CoV-2 virus	518	29.4
	reducing the risk for patients of contracting SARS-CoV-2 virus	500	28.4
	quick contact with the patient	318	18.0
	possibility of providing medical advice to more patients at the same time	312	17.7
	easy handling of administrative matters, correction of chronic treatment, consultation of results, or extension of prescriptions	58	3.3
	the possibility of better preparation for the visit	10	0.6
	enabling the division of patients into infectious and non-infectious	7	0.4
	better control of chronic disease therapy	5	0.3
	other (e.g., the convenience of patients who do not have to leave their home)	35	2.0
	total number of responses	1763	100.0
Disadvantages of teleconsultation	no possibility of personal examination of the patient	649	34.0
	unreliable transmission of information	428	22.4
	difficulties resulting from the patient’s symptoms (e.g., deterioration of hearing, psychotics disorders)	430	22.5
	technical difficulties	372	19.5
	long-winded conversations due to patients’ chattering or need to explain slowly various health issues	12	0.6
	patients are unprepared for teleconsultation, problems with calling the patient	10	0.5
	lack of non-verbal communication	4	0.2
	bad attitude of elderly patients to teleconsultations	2	0.1
	no disadvantages	2	0.1
	total number of responses	1909	100.0

**Table 4 ijerph-19-11532-t004:** Evaluation of the effectiveness of teleconsultation according to the age, profession, and medical specialization of the respondents.

Variable	Categories	No.	n	M	SD	Me	Wilcoxon Matched-Pairs Test						
age	20–30 years	(1)	123	6.48	2.15	7	* **p** *	1	2	3	4		
	31–40 years	(2)	238	6.42	2.22	7	2	1					
	41–50 years	(3)	181	6.18	2.34	6	3	1	1				
	51–60 years	(4)	165	5.94	2.27	6	4	0.299	0.165	1			
	>60 years of age	(5)	58	5.38	2.54	5	5	0.025	0.019	0.160	0.536		
profession	doctor	(1)	544	6.79	1.97	7	* **p** *	1	2	3	4		
	nurse	(2)	141	4.96	2.23	5	2	0					
	physiotherapist	(3)	27	3.33	2.60	3	3	0	0.008				
	paramedic	(4)	24	4.17	2.28	4	4	0	0.444	0.444			
	midwife	(5)	28	5.32	2.39	5	5	0.008	0.696	0.026	0.444		
medical specialization *	family medicine	(1)	260	7.27	1.63	7	* **p** *	1	2	3	4	5	6
	internal diseases	(2)	52	5.69	2.32	6	2	0					
	pediatrics	(3)	27	6.37	2.02	7	3	0.643	1				
	cardiology	(4)	23	5.61	2.21	5	4	0.003	1	1			
	hematology	(5)	10	5.20	2.20	6	5	0.059	1	1	1		
	anesthesiology	(6)	11	4.91	2.70	4	6	0.074	1	1	1	1	
	other	(7)	73	5.27	2.37	5	7	0	1	0.369	1	1	1

Note. *p*—calculated significance level in Wilcoxon matched-pairs test, *p* = 0 means that *p* < 0.001; * the group of respondents was limited to those who indicated only one specialization.

**Table 5 ijerph-19-11532-t005:** Evaluation of the reliability of teleconsultation according to the age, profession, and medical specialization of the respondents.

Variable	Categories	No.	n	M	SD	Me	Wilcoxon Matched-Pairs Test						
age	20–30 years	(1)	123	6.10	2.15	6	** *p* **	1	2	3	4		
	31–40 years	(2)	239	6.01	2.26	6	2	1					
	41–50 years	(3)	181	5.84	2.29	6	3	1	1				
	51–60 years	(4)	165	5.81	2.16	6	4	1	1	1			
	>60 years of age	(5)	58	5.34	2.60	5	5	0.770	0.810	1	1		
profession	doctor	(1)	544	6.49	1.97	7	** *p* **	1	2	3	4		
	nurse	(2)	142	4.59	2.13	5	2	0					
	physiotherapist	(3)	27	3.22	2.12	3	3	0	0.018				
	paramedic	(4)	24	4.00	2.45	3	4	0	0.343	0.681			
	midwife	(5)	28	5.11	2.47	4	5	0.008	0.713	0.045	0.343		
medical specialization *	family medicine	(1)	260	6.97	1.67	7	** *p* **	1	2	3	4	5	6
	internal diseases	(2)	52	5.33	2.17	5	2	0					
	pediatrics	(3)	27	5.48	2.05	5	3	0.008	1				
	cardiology	(4)	23	5.48	2.17	5	4	0.006	1	1			
	hematology	(5)	10	5.00	1.70	6	5	0.020	1	1	1		
	anesthesiology	(6)	11	4.55	2.58	4	6	0.027	1	1	1	1	
	other	(7)	74	4.93	2.35	5	7	0	1	1	1	1	1

Note. *p*—calculated significance level in Wilcoxon matched-pairs test, *p* = 0 means that *p* < 0.001; * the group of respondents was limited to those who indicated only one specialization.

**Table 6 ijerph-19-11532-t006:** Willingness to use teleconsultation in everyday work after the end of the COVID-19 pandemic according to age, profession, and sex.

		Willingness to Use Teleconsultation in Everyday Work after the End of the COVID-19 Pandemic								Fisher’s TestCramer’s V
		Often		Occasionally		Never		No Opinion		
Variable	Categories	n	%	n	%	n	%	n	%	*p*
age	20–30 years	44	35.8	63	51.2	11	8.9	5	4.1	0.020 0.093
	31–40 years	71	30.0	125	52.7	30	12.7	11	4.6	
	41–50 years	38	21.1	98	54.4	34	18.9	10	5.6	
	51–60 years	39	23.8	91	55.5	30	18.3	4	2.4	
	>60 years	10	17.2	30	51.7	16	27.6	2	3.4	
sex	female	137	24.3	315	55.9	89	15.8	23	4.1	0.083 0.093
	male	65	32.8	92	46.5	32	16.2	9	4.5	
profession	doctor	189	34.8	307	56.5	37	6.8	10	1.8	0 0.312
	nurse	11	7.9	75	53.6	47	33.6	7	5.0	
	physiotherapist	0	0.0	5	19.2	16	61.5	5	19.2	
	paramedic	2	8.3	6	25.0	12	50.0	4	16.7	
	midwife	0	0.0	13	46.4	9	32.1	6	21.4	

Note. Cramers’ V—a coefficient determining the level of dependence between two nominal variables; *p*—calculated significance level in Wilcoxon matched-pairs test, *p* = 0 means that *p* < 0.001.

## Data Availability

The data presented in this study are openly available in FigShare at https://doi.org/10.6084/m9.figshare.20375526.v1 (accessed on 26 July 2022).

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
