# Peer review of "Perception and Attitude toward Teleconsultations among Different Healthcare Professionals in the Era of the COVID-19 Pandemic"

_ijerph, 2022, doi:10.3390/ijerph191811532_

Round 1

Reviewer 1 Report

Dear authors,

thank you very much for the opportunity to read your work. My opinion is, that the topic of teleconsultations is important to communicate, especially in the context of COVID pandemic, to nourish further discussion among the experts and especially the users among HCPs.

I appreciate that the authors are not promising any extraordinary complicated statistical tests without the proper design of hypotheses. Instead, they are using descriptive interpretation, which is perfectly suitable in the presented case.

The authors designed their own questionnaire. A main limiting factor of the questionnaire is the lack of any questions regarding technical proficiency/eHealth literacy. It would be interesting to see whether the participants used any eHealth apps (not only teleconsultations) before or whether they, for example, care about the security of the patient's data. Also, the terms of effectiveness or reliability could be elaborated in deeper detail, as in the current form they might cause some level of "granularity" based on the different understanding among the participants. The authors could possibly inspire themselves by the UTAUT model or other UX assessment techniques.

In the line 115 authors are mentioning that 412 responses were collected using an electronic version which equals to return of 9.48 % from the total of 4345 electronic invitations. Authors should specify in the brackets that it is 52.8 % of all included responses (both paper and electronic) after the exclusion of - specify the number of excluded participants. It is not clear from the text unless the reader sees Fig. 1 or reads section 3 Results.

Regarding the presentation of the figures, Fig. 1 needs to be redesigned due to some inconsistencies. The second row shows 780 participants in total, then 831 participants completed the questionnaire (third row), in the next step 31 were excluded (making 800 of them), however, then again, 780 participants were included. What happened to 20 of the participants? Please clarify and adjust the Figure correspondingly.

In Table 6 please correct "foctor" in the profession list.

The discussion section reminds rather a narrative review. Some of the aspects were not tested via the questionnaire - as the already mentioned technical proficiency (lines 305, 367, and 368). On lines 332 and 333 medical examinations are mentioned. In fact, there were no questions regarding the use of point-of-care devices transmitting the data to the HCPs within the questionnaire. It could be also interesting to ask within the questionnaire whether the participants are aware of possibilities of the state-of-the-art telemetry and whether they actively use it (for instance for the teleauscultation, blood pressure measurement etc.). Also, the age of the participants is presented as a proxy for eHealth literacy. This is also limiting due to the fact, that the participants were not directly asked about their proficiency and due to the size of the aging physicians cohort. Limitations of the study in the terms of the eHealth literacy assessment should be stressed even more.

The article can be published after the aforementioned points are corrected/improved/clarified.

Reviewer 2 Report

The Paper is interesting.

The Covid era gave the opportunity to use telemedicine in medical practice; this tool could be useful and capable of optimizing medical assistance; nonetheless, there are problems with safety and medicolegal aspects that I think the authors have to face up.  It could be helpful that in the discussion, the authors talk about the medical liability that can be related to this new approach evaluating, for example: in which case this tool can be safer and when it's not safe. Another aspect that I think it's important is the application of specific procedures to set what kind of patient can be assisted with the telemedicine approach and in which part of the care process this tool can be used. On these aspects, I suggest seeing this article that talks about some of these issues: "Telemedicine as a medical examination tool during the Covid-19 emergency: The experience of the onco-haematology center of tor vergata hospital in Rome" DOI: 10.3390/ijerph17238834 This article evaluated the effect of telemedicine in the onco-haematology follow-up of patients, with risk and benefit related with this tool.  

Reviewer 3 Report

I appreciate the work that went into this study. However, there are some areas that need some consideration and revision. Please consider the following suggestions and perhaps some queries also to strengthen this manuscript.

This article reports the Perception and Attitude towards Teleconsultations among 2 Different Health Care Professionals in the Era of the COVID-19 Pandemic

The authors did very good effort to write a scientific sound paper with relevant references though the described study was based on a too limited and not homogeneous study sample.

In addition,

1-          Sample size: there was no sample size justification and no power calculation.

2-          The questionnaire is not validated to meet the criteria of a sound study of research. My advice is to improve your analysis of 11 questions concerning various aspects of the work of HCPs because it is confused. 

It is not clear why the author used Fisher Exact Test. Please explain them.

In addition, please explain this questionnaire has been developed for this purpose, and why the authors did not use any other validated scale for these specific questions. I think this needs to be justified in order to provide a sound rationale for using this particular questionnaire
